# The High Efficiency of Anionic Dye Removal Using Ce-Al_13_/Pillared Clay from Darbandikhan Natural Clay

**DOI:** 10.3390/molecules24152720

**Published:** 2019-07-26

**Authors:** Bakhtyar K. Aziz, Dler M. Salh, Stephan Kaufhold, Pieter Bertier

**Affiliations:** 1College of Medicals and Applied Sciences, Charmo University, Chamchamal 46023, Iraq; 2Clay and Environmental Chemistry Research Group, Department of Chemistry, University of Sulaimani, Sulaimaniyah 46001, Iraq; 3Federal Institute for Geosciences and Natural Resources (BGR), Stilleweg 2, 30655 Hannover, Germany; 4Dynchem Scientific Instruments, 52066 Aachen, Germany

**Keywords:** pillared clay, acid activation, anionic dye, isotherms, kinetics, thermodynamics

## Abstract

Natural clay from Darbandikhan (DC) was evaluated in its natural form, after acid activation (ADC), and after pillaring (PILDC) as a potential adsorbent for the adsorption of methyl orange (MO) as a model anionic dye adsorbate. The effect of different clay treatments was investigated using X-Ray Diffraction (XRD), X-Ray Fluorescence (XRF), Scanning Electron Microscope (SEM) and Fourier-Transform Infrared Spectroscopy (FT-IR), and N_2_ physisorption analysis. Both acid activation and pillaring resulted in a significant increase in adsorption affinity, respectively. The adsorption favored acidic pH for the anionic dye (MO). The adsorption process was found to follow pseudo-second-order kinetics with activation energies of 5.9 and 40.1 kJ·mol^−1^ for the adsorption of MO on ADC and PILDC, respectively, which are characteristic of physical adsorption. The adsorption isotherms (Langmuir, Redlich-Peterson and Freundlich) were fitted well to the experimental data. The specific surface area of the natural clay was very low (22.4 m^2^·g^−1^) compared to high-class adsorbent materials. This value was increased to 53.2 m^2^·g^−1^ by the pillaring process. Nevertheless, because of its local availability, the activated materials may be useful for the cleaning of local industrial wastewaters.

## 1. Introduction

Urban and industrial wastewater of the Sulaimani governorate is introduced into Darbandikhan Dam Lake through the Tanjero River. The polluted Tanjero river contains colored compounds, heavy metal compounds, and pharmaceutical wastes from a pharmaceutical company and many clinical labs. The hazardous character of these colored compounds results from their high toxicity and resistance to biodegradation [1]. Among the several methods used in the treatment of dye-contaminated effluents, the adsorption method is common because of its low cost, ease of operation, and effectiveness [2]. Sorption may be used not only for the removal of dyes, but also for a big amount of pollutants and proteins. Activated carbon, silicate materials, biomasses [3] and chitosan [4] are common efficient and low-cost adsorbents [5]. The cost/efficiency ratio determines the manufacturing procedure of these adsorbents [6]. For the development of adsorbents for the clarification of industrial wastewaters, local clays are often used to minimize the costs. Their lower adsorption capacity compared to high-class adsorbents can be compensated by increasing the amount of the available cheap material. 

Different methods exist to improve the specific material properties, i.e., the surface acidity or porosity, of these clays [7]. To improve the adsorption capacity, acid activation [8], base activation [9], intercalation [10], and pillaring [11] have been used.

Acid activation and pillaring are known to increase the surface area and, hence, are deemed to improve dye adsorption capacity. Acid activation results in the partial dissolution of the octahedral sheet and leaves some mesoporous silica behind [12]. For the pillaring of swellable clay minerals, large inorganic molecules are intercalated. Subsequent calcination leads to the development of oxidic species between the smectite layers which, in turn, results in an open and inflexible structure [13].

Improving the thermal stability, adsorptive and catalytic properties of pillared clay are very important. One of the improving methods was to incorporate different mixed-oxides to produce pillared clays with enhanced properties; mostly Al-pillared clays in which other metal cations are introduced [14,15]. Normally, montmorillonites-rich clays that have a large surface area and swelling capacity are used as a precursor for the preparation of pillared clays.

Natural clays are effective, cheap and abundant adsorbents, but show poor adsorption efficiency toward anionic dyes [16]. Some clay minerals and their acid-treated or modified forms have been recognized for their adsorption capacity for methyl orange (MO), such as Moroccan natural clays [17], heat-treated palygorskite clays [18], intercalation of cetyltrimethylammonium bromide to vermiculite interlayer spacing [19], and protonated cross-linked chitosan [20]. Methyl orange is widely used as a model anionic dye in adsorption and photocatalytic degradation studies.

The adsorption capacity for MO is considerably affected by the surface functional groups and surface charge of the adsorbents, as well as by the pH and temperature of the medium. The values of these parameters point to the possible involvement of physical forces such as hydrogen bonding, Vander Waals and covalent chemical bonds in the adsorption process.

In the presented study, methyl orange is used as a model substance for actual wastewaters. The aim of this research was to characterize a natural and activated local clay and assess its applicability from adsorption tests with methyl-orange as a model anionic dye. 

## 2. Materials and Methods

### 2.1. Adsorbate

A model anionic dye, methyl orange (MO) (Merck, Germany), was used as the adsorbent (λ_max_ at pH ≥ 3.1 = 507 nm, and at pH < 4.4 = 618 nm). The concentrations of the dye solutions ranged from 6 to 500 mg/L by dilution from a 500 mg/L stock solution of MO using distilled water. Methyl orange is present in an alkaline medium in the monomer form for concentrations ≤ 1.2 × 10^−4^ mol/L. In 0.032 to 0.46 mol/L HCl (Merck, Germany), it is present as a tautomeric mixture. The pKa value for methyl orange is 3.37 ± 0.01 [21]. 

### 2.2. Adsorbents

Three adsorbents were used in this study, namely, the natural clay of Darbandikhan (DC), its acid-activated form (ADC), and Ce-Al/pillared clay (PILDC) prepared from the natural clay of Darbandikhan. The natural clay was fractioned by successive dilution and sedimentation in 1 L cylinders. The <5 µm fraction of the clay was separated by sedimentation and used in the present study.

#### 2.2.1. Acid-Activated Clay

The natural clay (DC) was mixed with 2 M HCl at a solid-to-liquid ratio of 2:10 w/v in a 1 L conical flask and stirred under atmospheric pressure using a digitally controlled stirrer for 3 h at 70 °C. The remaining solid was collected by centrifugation for 10 min (5000 rpm) at a relative centrifugal force (RCF) = 4193 using the Universal 320 centrifuge from Hettich GmbH & Co. KG (Tuttlingen, Germany), and the residue was rinsed five times with distilled water to remove the residual acid on the surface of the sample. Finally, the residue was dried overnight at 100 °C and sieved to pass a 200 µm sieve.

#### 2.2.2. Ce-Al_13_/Pillared Clay

The pillaring solution was prepared by the slow addition of a sodium hydroxide solution (0.2 M NaOH) (2–3 mL/min.) to a 0.2 M solution of AlCl_3_ until a ratio of 1:1 was achieved at which pH ≈ 4. The mixture was allowed to stand at 70 °C overnight with constant stirring. A total of 5 mg of cerium nitrate was dissolved in 100 mL distilled water and the solution was introduced to the pillaring solution and allowed to stand for 6 h at 70 °C with constant stirring.

The clay was Na-saturated by mixing 30 g of the natural clay with 1 L of 2 M NaCl solution and stirring for 3 h, then washed with distilled water until chloride could not be detected anymore, and finally dried at 100 °C overnight. The Na-form of the clay was dispersed in 200 mL of distilled water at 60 °C by prolonged stirring (5 h) on a magnetic stirrer. The amount of pillaring solution required to obtain a [Ce-Al/Clay] ratio of 20 mmol Al/g clay was then added to the vigorously stirred dispersion. The final suspension was allowed to stand for 5 h. The solid material was separated by centrifugation and washed with distilled water until the supernatant was free of chloride which was detected using a 0.1 M AgNO_3_ solution [22,23]. The dried powder was directly used in the experiments without further treatment.

### 2.3. Adsorbent Characterization

The chemical composition of DC was determined by XRF. For the XRF analysis of powdered samples, a PANalytical Axios spectrometer (Almelo, The Netherlands) was used. The clay sample (DC) was mixed with lithium metaborate flux and melted to prepare the sample glass beads. The beads were analyzed by wavelength-dispersive XRF. A total of 1000 mg of sample material was heated to 1030 °C for 10 min to determine the loss on ignition (LOI).

All materials were investigated by XRD. XRD patterns were recorded using a PANalytical X’Pert PRO MPD Θ-Θ diffractometer (Almelo, The Netherlands), (Cu-Kα radiation generated at 40 kV and 30 mA), equipped with a variable divergence slit (20-mm irradiated length), primary and secondary soller and Scientific X’Celerator detector (active length 0.59°). With a step size of 0.0167° 2Θ, the samples were investigated from 2° to 85° 2Θ with a measuring time of 10 sec per step. The top-loading technique was used for specimen preparation. A detailed clay mineralogical investigation was performed on the textured slides of the <2 µm fraction. A total of 15 mg per cm^2^ clay was used. An aliquot of 1.5 mL of suspension was deposited on circular (diameter = 2.4 cm) porous ceramic tiles that were 3 mm thick, mounted on a vacuum filtration apparatus. Furthermore, the specimens were stored overnight in an ethylene glycol atmosphere at 60 °C. The clay films were measured from 1° to 40° 2Θ (step size 0.03° 2Θ, 5 s per step) after cooling to room temperature, representing EG conditions.

Mid-infrared spectra were measured on a Thermo Nicolet Nexus FTIR spectrometer (Thermo Electron Scientific Cor., Madison, WI, USA) (MIR beam splitter: KBr, detector DTGS TEC) with the KBr pellet technique (1 mg sample/200 mg KBr). The resolution was adjusted to 2 cm^−1^. Measurements were conducted before and after drying of the pellets at 150 °C in a vacuum oven for 24 h.

Thermal gravimetric analyses (TGA) for the samples were performed on a Diamond Thermal Gravimetric—Diffrential Thermal Analysis (TG-DTA) (SII) thermogravimetric analyzer from PerkinElmer (Shelton, CT, USA) using a 50 mL/min flow rate of N_2_ inert gas from room temperature up to 1000 °C at a rate of 20 °C/min.

N_2_ physisorption isotherms were measured on a Micrometrics Gemini VII 2390t apparatus (Micromeritics Instrument Corporation, Norcross, GA, USA). Adsorption and desorption measurements were performed in a cryogenic nitrogen bath (77 K) at 99 discrete pressure points between 0.001 and 0.995 p/p°. The saturation pressure (p°) was recorded independently for every pressure step. Sorption equilibrium was assumed when the pressure variation per equilibration time (10–40 s) was less than 0.01%. The samples were outgassed at 250 °C to a residual pressure below 10 mPa. The specific surface area (SBET) was determined by applying the BET model to the N_2_ adsorption data in the p/p_o_ ranges as recommended by Rouquerol et al. (2007) [24]. The specific total pore volume (VT) was calculated according to Gurvich’s rule, at the highest relative pressure measured and reported with the corresponding BJH pore radius [25]. The specific micropore volume (Vp) was determined using the Dubinin equation. The cation exchange capacity (CEC) was measured using the Cu-Triethylenetetramine method [26].

### 2.4. Adsorption Studies

Adsorption investigations were performed as batch experiments. In all experiments, 0.1 g of adsorbent was added to a 120 mL polyethylene bottles containing 50 mL of the MO dye solution with variable concentrations. The dye-clay mixture was agitated at a rate of 200 rpm in a thermostatic water bath shaker. After adsorption, the dispersions were centrifuged for 5 min at a rate of 4500 rpm. The residual concentrations of MO in the supernatants were analyzed by standard calibration using a UV–Vis spectrophotometer at (λ_max_ = 618 nm). The amount of adsorbed dye, q_e_ (mg/g), was calculated using the relationship in Equation (1):(1)$$q_{e}=\;\frac{\left(C_{o}-C_{e}\right)\;V}{m}$$
where C_o_ and C_e_ are the initial and equilibrium concentrations of the dye (mg/L), respectively, V is the volume of the solution (L), and m is the mass of the adsorbent (g).

The effect of the equilibration time was investigated by varying the contact time from 0 to 400 min at room temperature. The effect of the initial pH of the dye solution was studied over a range of 2.0 to 9.0. The pH values of the solutions were adjusted using the HCl (0.1 mol/L) and NaOH solutions (0.1 mol/L) and monitored with a pH meter. To study the effect of the initial dye concentration, the experiments were performed at various initial dye concentrations ranging from 6 to 80 mg/L (6, 10, 20, 30, 50 and 80 mg/L).

The Langmuir model describes monolayer (one molecule in thickness) adsorption, assuming the existence of a fixed number of identical and equivalent active sites on the adsorbent surface. The graph of the Langmuir isotherm is characterized by a plateau [27] and it is expressed as in Equation (2):(2)$$q_{e}=\frac{q_{m}K_{L}C_{e}}{1+K_{L}C_{e}}$$
where q_e_ (mg·g^−1^) is the MO equilibrium adsorption capacity, q_m_ (mg·g^−1^) is the maximum adsorption capacity, K_L_ (L·mg^−1^) is the Langmuir equilibrium constant and C_e_ is the concentration of the adsorbent.

The Freundlich isotherm has been used mostly to describe the sorption of organic chemicals on soils [28]. The underlying assumption is that adsorption sites are not equivalent and, hence, adsorption occurs first on stronger binding sites (Equation (3)) [29].
(3)qe=KfCe1n
where n and K_F_ are the Freundlich constants.

The Redlich-Peterson (R-P) isotherm (Equation (4)) was applied as a three-parameter isotherm using non-linear curve fitting with the aid of the OriginPro (2017) computer software with the Levenberg Marquardt iteration algorithm.
(4)$$q_{e}=\frac{K_{RP}C_{e}}{1+\;a_{RP}C_{e}^{\beta }}$$
where K_RP_ (L/mg) and α_RP_ (L/mg) are the Redlich-Peterson constants. The exponent β lies between 0 and 1, where the Redlich–Peterson equation becomes the Langmuir equation when β = 1, and it becomes Henry’s law when β = 0

Three error functions were applied to determine the goodness of the curve fittings for the kinetic and the isotherm experimental data. The error functions were (the correlation coefficient (R^2^), the sum of squared errors (SSE) and the deviation (Δq) of the calculated adsorption capacity (q_calc_) from the experimental one (q_exp_)) as follows:(5)$${\;R}^{2}=1-\;\frac{\sum {\left(q_{\exp}-q_{\mathrm{fit}}\right)}^{2}}{\sum {\left(q_{\exp}-q_{\mathrm{mean}}\right)}^{2}}$$
(6)$$\mathrm{SSE}=\;\sum {\left(q_{\exp}-{\;q}_{\mathrm{fit}}\right)}^{2}$$
(7)$$\Delta q=\;\frac{q_{\exp}-{\;q}_{\mathrm{calc}}}{q_{\exp}}\;\times 100$$

## 3. Results and Discussion

### 3.1. Characterization of the Adsorbent

The X-ray fluorescence (XRF) method has been utilized to recognize the major minerals and chemical compounds present in the clays [30]. The chemical compositions of DC and its derivatives (ADC and PILDC) are presented in Table 1.

The calcite content of DC has been reduced to a great extent from 21.4 CaCO_3_ (12.0% CaO) to 2.3% and 0.7% CaCO_3_ (1.3% and 0.4% CaO) for ADC and PILDC, respectively, by the action of the acid activation (dissolution) (Table 1) in addition to an increase in the porosity of ADC. The relative enrichment of SiO_2_ and Al_2_O_3_ may be related to the decrease in the calcite content for ADC and PILDC. In the case of the PILDC, the ratio of the Al_2_O_3_ increased due to the insertion of the pillaring agent in addition to the effect of acid dissolution.

The XRD patterns (Figure 1a) show that the precursor material DC contains a 14.3 Å mineral (smectite, vermiculite, chlorite), some illite (10 Å), quartz, feldspar, and calcite. According to the chemical composition, about 20 mass% calcite is present in this sample. The CEC of the clay sample was 13 meq/100 g which points towards a low smectite content of 10–15 mass%. Acid treatment resulted in the dissolution of calcite in both treated samples, ADC and PILDC. For the pillared material, some intensity was found between 20 and 30 Å, which corresponds to the permanently expanded smectite layers due to the produced pillars. The intensity, however, was weak most probably because of the low smectite content. The acid-activated material showed no sign of decreasing the intensity of the clay minerals. As indicated by the chemical analysis, acid activation mostly resulted in the dissolution of carbonates rather than the activation of clay minerals.

The FTIR spectra of DC, ADC and PILDC are shown in (Figure 1b). The band at 3614 cm^−1^ could be assigned to the OH stretching of Chlorite + illite/smectite (dioctahedral) and the band at 3420 cm^−1^ was assigned to Fe-rich chlorite because no kaolinite was found (absence of 3700 cm^−1^), the XRD intensity at 7 Å, therefore, can be explained by the Fe-rich chlorite with a high intensity ratio of 002/001. The bands observed at 2876.10, 1432, and 712.87 cm^−1^. correspond to carbonate. After acid activation and pillaring, no carbonate bands were found anymore, which is in accordance with the XRD.

The thermal gravimetric analysis (TG) profiles of DC, ADC and PILDC are shown in Figure 1c. The first step of weight loss (30 to 200 °C), which corresponds to the loss of adsorbed and interlayer water [31], increased from 1.8 to 2.3% for DC and its acid-activated form ADC, respectively. This may be due to the increased porosity caused by the acid treatment or simply by the relative enrichment of clay minerals caused by the dissolution of carbonates. The adsorbed and interlayer water of PILDC has increased to 2.6% possibly due to the expansion of the interlayer by the pillaring process. For PILDC, there is a gradual weight loss between 200 to 450 °C which may result from the dehydroxylation of Ce-Al polyoxocations [32,33]. Between 460–590 °C, where the weight loss belongs to the dehydroxylation of the clay layer, there was no significant change in weight loss (1.88, 2.24 and 2.38% for DC, ADC and PILDC, respectively). Natural clay shows a mass loss of 9.35% at 620–820 °C which is due to the calcite content of the natural clay [34]. This region of weight loss disappeared for the acid-activated and pillared clays. The 9.35% weight loss of CO_2_ corresponds to the total calcite content of the clay (21.3%) which is consistent with the presence of 12.0% of CaO (Table 1). The 12.0% of CaO corresponds to about a 21.4% calcite content.

The adsorption-desorption isotherms of N_2_ gas at 77 K are shown in Figure 1d and the results of surface area, pore volume, and pore size measurements are summarized in Table 2. The specific surface area (S_BET_) of DC clay is 22.4 m^2^·g^−1^. The S_BET_ of ADC and PILDC increased with respect to the DC clay (26.3 and 53.2 m^2^·g^−1^). Acid activation resulted in a slight increase (4 m^2^·g^−1^) of the surface area which is related to an increase in microporosity [35]. The introduction of pillars resulted in a significantly larger microporosity. Compared to other pillared clays such as Bentonite [36] or acid-activated clays [37], however, the increase of the surface area was relatively low, which could be explained by the lower smectite content. The isotherm of the raw material (DC) displays low-pressure hysteresis, i.e., the desorption is above the adsorption branch at P/P_o_ below about 0.4. Such sorption behavior has been attributed to slow diffusion in pore networks in which significant meso- and macropore volume is accessible only through micropores [25]. This implies that the reported BET area is an underestimation of the true surface area and that the difference in the BET area between DC and ADC is probably insignificant. The hysteresis loop of DC is substantially wider than for ADC at high relative pressures, which indicates that larger mesopores and macropores with diameters below 300 nm are more prominent in raw rather than acid-activated material. The narrow hysteresis loops of ADC and PILDC suggest a disaggregation of the material which would be consistent with the loss of meso- and macroporosity.

### 3.2. Adsorption Studies

For the adsorption of an adsorbate on an adsorbent, the surface characteristics of the adsorbent, the chemistry of the adsorbate and the factors that affect those properties, such as pH and temperature, are important.

Preliminary studies on the adsorption of model anionic dye MO on the natural clay (DC) showed no detectable adsorption capability, therefore, attempts were made to achieve a good adsorption capacity for the natural clay using acid activation (ADC) and pillaring (PILDC) methods. 

#### 3.2.1. Operational Condition Effects

The impact of equilibrium time on the MO adsorption by the adsorbents (ADC and PILDC) is shown in Figure 2a. At the initial stage of adsorption, the amount of MO adsorbed increased rapidly, then slowed down until it reached equilibrium. The quick adsorption can be credited to a large number of empty adsorbent sites and the high gradient of the solute concentration in the early stages of adsorption. Figure 2a shows that 100 and 180 min were required to reach equilibrium for the adsorption of MO on ADC and PILDC, respectively. These conditions were used consistently throughout the rest of this study. 

The pH of the suspension has an important role in the adsorption process. It affects both the charge of the adsorbate as well as the charge on the adsorbent surface. The effect of the initial pH on the adsorption of MO on ADC and PILDC are shown in Figure 2b, within a pH range of 2.0 to 9.0. The adsorption capacity increased from pH 2.0 to 3.0, after which the adsorption capacity decreased to zero adsorption of MO at a pH above 5.0. At low pH values of the mixture, the surface and the active sites of the adsorbent were probably positively charged and, hence, electrostatic attraction occurs with the negatively charged MO. Increasing the pH will result in a decreasing number of positive charges at the adsorbent surface. At basic pH, two factors may contribute to the decrease of the adsorption capacity of the adsorbent; the competition influence of OH^-^ and the repulsive forces between the negatively charged surface and negative charge MO [38]. Consequently, the strong electrostatic interaction between the MO anion and the positive adsorbent surface could be the mechanism of the adsorption in acidic media [39].

The effect of the amount of adsorbent was studied in the range between 1–4 g L^−1^ using 20 mg·L^−1^ of MO as the initial concentration. Figure 2c shows that the percent of MO removal increased with the increase in both the adsorbent ADC and PDC. The removal percent of the ADC and PILDC increased from 20 to 47% and from 35 to 89%, respectively. The higher removal percent of the PILDC may be attributed to the increase of the surface acidity of the clay due to the pillaring process [40]. Increasing the adsorbent dose results in the increase of the total number of the active sites of the adsorbent and, hence, a higher removal percent of MO [19]. However, the increase in the removal efficiency was not linear with the increasing adsorbent dose which may be attributed to the aggregation of the adsorbent particles at higher doses [41].

The influence of the initial MO concentrations on the removal percentage and the pseudo-second-order kinetics (k_2_) at 40 °C and pH = 3.0 was studied in the range of 6–80 mg·L^−1^ dye concentration for both ADC and PILDC (Figure 2d). At low initial concentrations of MO, the removal % was minimum (≈10%) for both ADC and PILDC and increased to 57 and 80% by increasing the initial dye concentration to 50 mg·L^−1^ for ADC and PILDC respectively where it became nearly constant for higher initial concentration (80mg·L^−1^). This behavior may be due to the availability of a large number of active sites of the adsorbent unoccupied by the MO molecules at low concentrations of MO which makes the driving force enhanced by increasing the initial MO concentration until the active sites on the adsorbent were fully occupied at C_o_ = 50 mg·L^−1^ [42]. Meanwhile, the rate constant k_2_ decreased very sharply by increasing the C_o_, then it comes to a constant for the higher C_o_ above 20 mg·L^−1^. This may also be attributed to the high ratio of the binding sites of the adsorbent per adsorbate molecules at low C_o_. This ratio decreases as the C_o_ increases due to the saturation of the binding sites, and the k_2_ decreases as well [43].

#### 3.2.2. Adsorption Kinetics

The MO amount adsorbed per unit mass q_t_ of ADC and PILDC showed a positive trend with adsorption time (Figure 3a,b). Higher adsorption capacities were noticed for the pillared clay than the acid-activated clay. The equilibrium time was attained within 100 and 180 min for ADC and PILDC, respectively. Hoo’s pseudo-second-order kinetic model (Equation (8)) was applied with better fitting to the practical adsorption data than the pseudo-first-order kinetic model of Lagergren (Equation (9)) for both of the modified clays (Table 3).
(8)$$q_{t}=\;\frac{k_{2}q_{e}^{2}t}{1+{\;k}_{2}q_{e}^{2}t}$$
(9)$$q_{t}=q_{e}\left(1-e^{-k_{1}t}\right)$$

The kinetic parameters were estimated from the kinetic plots using non-linear curve fitting (Figure 3a,b) and the values are listed in Table 3 for both ADC and PILDC using a 30 ppm initial concentration of MO and pH = 3.0 at different temperatures. The values of k_2_ increased with an increase in temperature. Increasing the temperature may enhance the diffusion rate of the adsorbate molecules from the bulk solution to the adsorbent surface [20,39].

The activation energy for the adsorption process was calculated from the slope of the Arrhenius plot (ln k_2_ vs 1/T). The magnitude of E_a_ for the adsorption process on ADC and PILDC adsorbents were 5.9 and 40.1 kJ·mol^−1^ respectively. These values fall in the range of physisorption [44] (Figure 3c). However, the distinction between physical and chemical adsorption is difficult because no sharp distinction exists and intermediate cases are present. 

#### 3.2.3. Adsorption Isotherms

Adsorption isotherms explain the relations between equilibrium concentrations of adsorbate in the solid phase q_e_, and in the liquid phase C_e_ at a constant temperature. Adsorption isotherms can be described by various mathematical models such as one parameter isotherm (Henry’s isotherm), two-parameter isotherm (Langmuir, Freundlich, Temkin, etc.) or three-parameter isotherm (Redlich-Peterson, Sips, etc.). The two-parameter isotherms are usually linearized to obtain the isotherm parameters and the best fit is evaluated from correlation coefficients (R^2^) and the sum of squared errors (SSE).

The adsorption of MO on ADC (Figure 4a,c) and PILDC (Figure 4b,d) at different temperatures, shows good fitting of the isotherm models of Langmuir, Redlich-Peterson and Freundlich as depicted by the error functions R^2^ and SSE (Table 4).

Table 4 shows that both treatments ADC and PILDC resulted in a better affinity of the clay towards MO in contrast to the natural un-treated clay DC where no detectable adsorption was observed. Therefore, they are not included in the graphs and the table. The Freundlich K_F_ values for the pillared clay are higher than those of the acid-activated clay at the same temperatures. This observation indicates that PILDC has a better adsorption intensity for MO adsorption than ADC. The Langmuir adsorption capacity q_m_ of PILDC (248.7 mg/g) is much higher than ADC (68.1 mg/g). The significant increase in the removal efficiency of the PILDC cannot be totally ascribed to the expansion of the interlayer space, the catalytic degradation of the dye by Ce-Al_13_ pillared clay may be involved [36]. Similar results of the high removal efficiency of thymol by Al_13_-pillared bentonite [45] were recorded.

#### 3.2.4. Adsorption Thermodynamics

The removal of MO on the acid-activated clay and pillared clay were studied at different temperatures (20, 30, 40 and 50 °C). The thermodynamic parameters—enthalpy change (ΔH°), entropy change (ΔS°) and Gibbs free energy change (ΔG°)—were calculated from the relations below:(10)$$\Delta G{}^{\circ}=\;-RT\ln K_{C}$$
(11)ΔG°= ΔH°−TΔS°
(12)$$\ln K_{C}=\;\frac{\Delta S{{}^{\circ}}^{}}{R}-\;\frac{\Delta H{{}^{\circ}}^{}}{RT}$$
where K_C_ is the thermodynamic equilibrium constant. The values of K_C_ were determined by plotting ln(q_e_/C_e_) vs. q_e_ (Figure 5a,b) [39,46]. ΔH° and ΔS° were evaluated from the slope and intercept of the linear plot of ln K_C_ vs 1/T as shown in Figure 5c, and the thermodynamic parameters are listed in Table 5:

An endothermic process (positive ΔH°) was recorded for the adsorption of MO on ADC and PILDC with an increase in the randomness at the adsorbent-solution interface, which is due to the desorption of the previously adsorbed water molecules on the surface of the adsorbent and the removal of most of the water of the solvation of the solvated MO as assigned by the positive values of ΔS° [39]. The observed ΔG° values were negative, pointing to the spontaneous and favorable adsorption of MO on both ADC and PILDC. 

It is necessary to compare the adsorption capacity of the current adsorbents with those reported in the literature. Although, consideration of economic aspects should be taken into account when selecting an adsorbent for wastewater treatment. Table 6 shows the Langmuir adsorption capacity q_m_ of ADC and PILDC from the present study compared to the efficient adsorbents from the literature. Compared to the efficient adsorbents, ADC is less efficient, while PILDC shows a superior adsorption capacity over most of the efficient adsorbents. 

## 4. Summary and Conclusions

In this study, natural clay of Darbandikhan (DC), its acid-activated form (ADC), and a pillared clay (PILDC) product were characterized and evaluated for MO adsorption of aqueous media. PILDC has been presented to be a highly efficient adsorbent for the removal of the anionic dye MO. The high removal efficiency was ascribed to the catalytic degradation of MO in addition to the adsorption process. The initial pH of the mixture had a significant effect on the anionic MO dye adsorption and the optimal pH was found to be 3.0 to observe the maximal adsorption capacity. This was correlated to the electrostatic interaction between surface active sites and the charged functional groups of the dye. Pseudo-second-order kinetics was found to govern the adsorption process. The experimental results suitably fit the Freundlich model. The thermodynamic data were based on adsorption tests conducted at pH 3.

Desorption, regeneration and column (continuous system) are to be considered in the future work to bring the current work to the applied world.

## Figures and Tables

**Figure 1 molecules-24-02720-f001:**
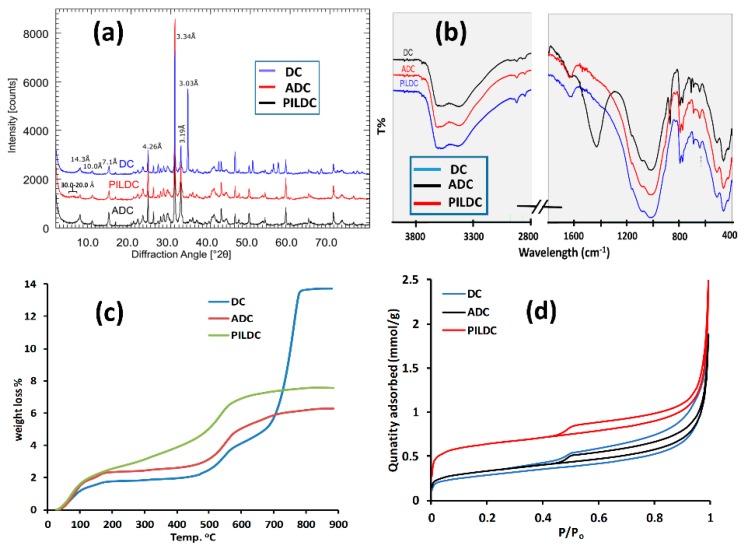
(**a**) The X-ray diffractogram for Darbandikhan (DC) (blue), after acid activation (ADC) (black) and after pillaring (PILDC) (red), (**b**) FTIR Spectra (**c**) TG curves for DC (blue), ADC (red), PILDC (green) and (**d**) Adsorption-desorption isotherms of N_2_ at 77 K for DC, ADC and PILDC.

**Figure 2 molecules-24-02720-f002:**
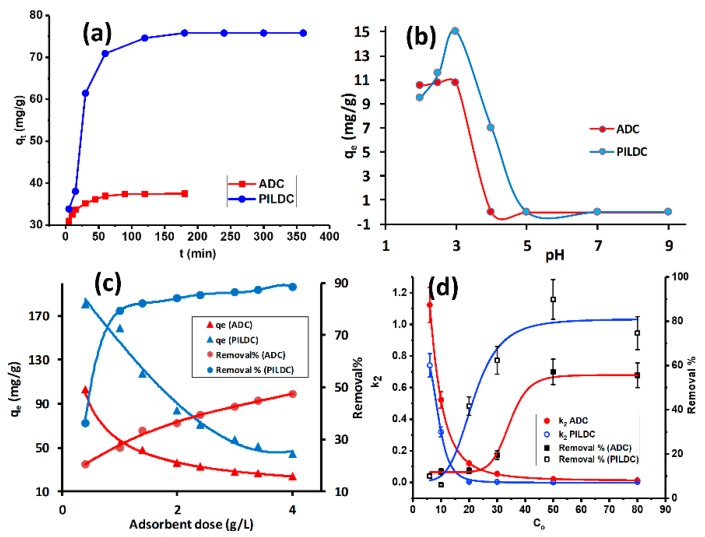
(**a**) The equilibrium time for the adsorption of MO on ADC and PILDC at 40 °C, (**b**) the effect of the initial pH of the MO solution on the adsorption on ADC and PILDC, (**c**) the effect of adsorbent dose on the adsorption efficiency of MO and (**d**) the effect of initial MO concentration on the removal % and k_2_ for ADC and PILDC at pH 3.0 (error bars correspond to 5% data error).

**Figure 3 molecules-24-02720-f003:**
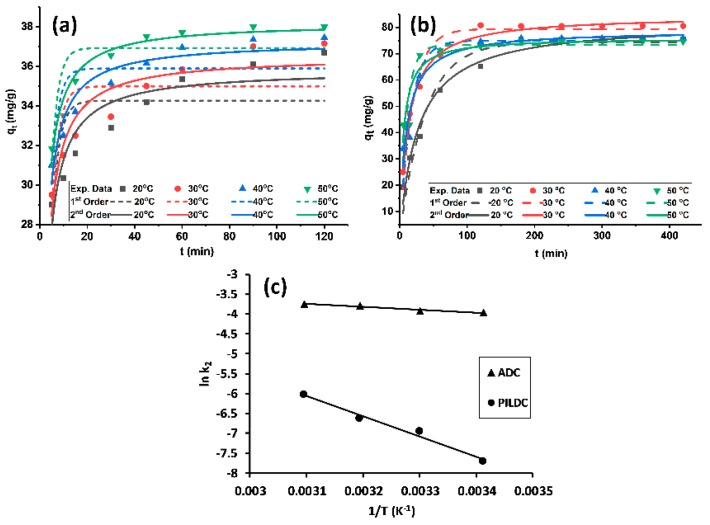
(**a**) Kinetic plots for the adsorption of MO on ADC, (**b**) PILDC and (**c**) Arrhenius plots for the adsorption of MO on ADC and PILDC at pH 3.0.

**Figure 4 molecules-24-02720-f004:**
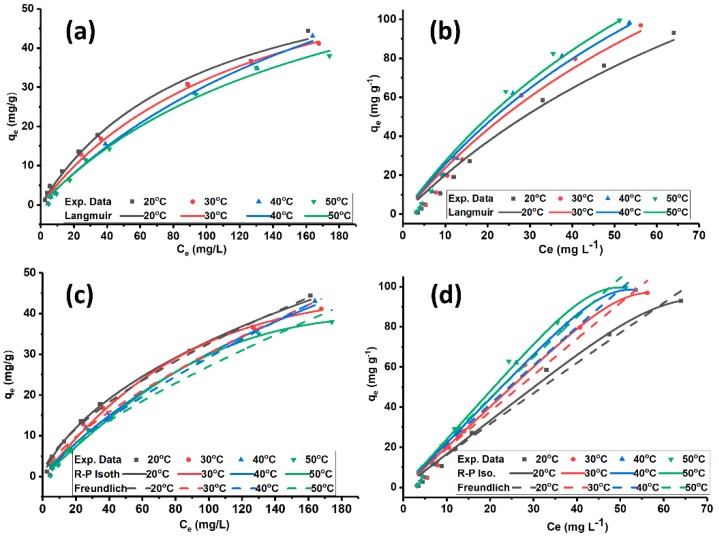
Langmuir adsorption isotherms of MO on (**a**) ADC and (**b**) PILDC, and Redlich Peterson and Freundlich adsorption isotherms of MO on (**c**) ADC and (**d**) PILDC at pH 3.0.

**Figure 5 molecules-24-02720-f005:**
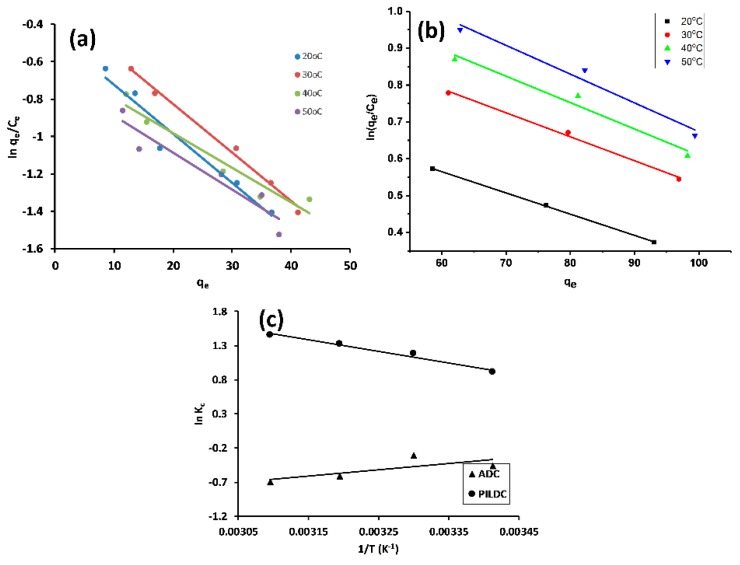
Khan plots for the adsorption of MO on ADC (**a**) PILDC (**b**) and Vant Hoff’s plot at pH 3.0 (**c**).

**Table 1 molecules-24-02720-t001:** The chemical compositions of the after acid activation (ADC), after pillaring (PILDC), and Darbandikhan (DC).

	SiO_2_%	TiO_2_%	Al_2_O_3_%	Fe_2_O_3_%	MnO%	MgO%	CaO%	Na_2_O%	K_2_O%	P_2_O_5_%	SO_3_%	LOI	Sum
DC	45.4	0.8	12.5	6.2	0.1	4.3	12.0	1.5	1.9	0.1	0.04	14.9	99.7
ADC	57.4	0.9	15.6	7.5	0.1	5.0	1.3	1.9	2.4	0.1	<0.01	7.5	99.7
PILDC	54.9	0.9	19.1	7.2	0.1	4.7	0.4	1.7	2.2	0.1	<0.01	8.6	99.01

**Table 2 molecules-24-02720-t002:** The N_2_ adsorption results obtained for DC, ADC and PILDC.

Material	*S_BET_* (m^2^·g^−1^)	t-Plot		Pore Volume		BJH Pore Diameter (nm)
		*S _micropore_* (m^2^·g^−1^)	*S_ext_* (m^2^·g^−1^)	*V_T_* (cm^3^·g^−1^)	t- Micro Volume (cm^3^·g^−1^)	
DC	22.39	9.72	12.68	0.0652	0.00503	13.91
ADC	26.33	6.53	19.80	0.0646	0.00290	15.78
PILDC	53.20	29.95	23.25	0.0877	0.01210	17.46

**Table 3 molecules-24-02720-t003:** The pseudo-second-order kinetic parameters for the adsorption of MO onto ADC and PILDC at pH = 3.0.

	Kinetics Model	Kinetic Parameters	Temperature (K)			
			293	303	313	323
**ADC**	**Experimental**	**q_exp_** **(mg/g)**	36.5	37.0	37.5	38.0
	**Pseudo-first-order**	**k_1_(min^−1^)**	0.3312	0.3347	0.3631	0.3640
		**q_calc_** **(mg/g)**	34.26	34.99	35.89	36.92
		**Dev. % of q_calc_**	6.1	5.4	4.3	2.8
		**SSE**	27.0	23.2	17.3	12.5
		**R^2^**	0.506	0.558	0.574	0.665
	**Pseudo-second-order**	**k_2_**	0.0191	0.0196	0.0226	0.0234
		**q_calc_**	35.83	36.5	37.3	38.2
		**Dev. % of q**	1.8	1.4	0.5	0.5
		**SSE**	7.9	6.5	3.4	1.5
		**R^2^**	0.855	0.876	0.915	0.960
**PILDC**	**Experimental**	**q_exp._** **(mg/g)**	74.5	78.6	77.0	77.0
	**Pseudo-first-order**	**k_2_**	0.0268	0.0528	0.0610	0.0968
		**q_calc_** **(mg/g)**	74.3	79.4	75.0	73.3
		**Dev. % of q_calc_**	0.3	1.0	2.6	4.8
		**SSE**	198.9	140.5	257.9	404.9
		**R^2^**	0.953	0.958	0.893	0.728
	**Pseudo-second-order**	**k_2_**	4.5 × 10^−4^	9.6 × 10^−4^	13.1 × 10^−4^	24.1 × 10^−4^
		**q_m_**	82.1	84.5	78.9	76.0
		**Dev. % of q_calc_**	10.2	7.5	2.5	1.3
		**SSE**	7.9	6.5	3.4	1.4
		**R^2^**	0.987	0.991	0.931	0.843

**Table 4 molecules-24-02720-t004:** Langmuir, R-P and Freundlich isotherm parameters for the adsorption of MO on ADC and PILDC.

Material	Isotherm	Temp. (K)	293	303	313	323
**ADC**	**Langmuir**	**K_L_ (L/mg)**	0.0102	0.0075	0.0044	0.0056
		**q_m_ (mg/g)**	68.1	75.0	100.0	79.8
		**SSE**	12.4	14.6	12.4	12.1
		**R^2^**	0.994	0.993	0.994	0.993
	**Redlich-Peterson**	**K_RP_ (L/mg)**	1.23	0.49	0.48	0.37
		**α_rp_ (L/mg)**	0.19	7.17	0.02	3 × 10^−5^
		**β**	0.58	1.42	0.79	1.93
		**SSE**	4.7	11.2	11.9	6.4
		**R^2^**	0.998	0.994	0.994	0.996
	**Freundlich**	**K_F_ (mg/g)(L/mg)^1/n^**	1.747	1.167	0.795	0.877
		**n**	1.57	1.42	1.28	1.34
		**SSE**	9.1	44.4	17.6	33.6
		**R^2^**	0.995	0.977	0.991	0.981
**PILDC**	**Langmuir**	**K_L_ (L/mg)**	0.0088	0.0098	0.0104	0.0107
		**q_m_ (mg/g)**	248.7	264.2	271.9	279.5
		**SSE**	241.2	308.7	298.5	308.4
		**R^2^**	0.975	0.970	0.972	0.972
	**Redlich-Peterson**	**K_RP_ (L/mg)**	1.66	2.01	2.21	2.40
		**α_rp_ × 10^12^ (L/mg)**	5.2	0.27	0.35	9.5
		**Β**	5.8	6.7	6.8	6.1
		**SSE**	96	135	136	154
		**R^2^**	0.990	0.987	0.987	0.986
	**Freundlich**	**K_F_ (mg/g)(L/mg)^1/n^**	1.844	2.193	2.628	2.979
		**n**	1.05	1.05	1.08	1.10
		**SSE**	134.8	202.2	238.5	277.4
		**R^2^**	0.984	0.978	0.975	0.971

**Table 5 molecules-24-02720-t005:** The thermodynamic data for the adsorption of MO on ADC and PILDC at pH 3.0.

		Acid-Activated Clay (ADC)				Pillared Clay (PILDC)		
Temp. (K)	Ln K_c_	ΔH° (kJ/mol)	ΔS° (J/mol)	ΔG° kJ/mol	Ln Kc	ΔH° (kJ/mol)	ΔS° (J/mol)	ΔG° kJ/mol
**293**	−0.458	7.86	29.86	−10.54	0.911	14.03	55.74	−2.30
**303**	−0.307			−11.17	1.180			−2.86
**313**	−0.695			−11.79	1.325			−3.42
**323**				−12.42	1.454			−3.97

**Table 6 molecules-24-02720-t006:** The comparison of adsorption capacities for methyl orange on efficient adsorbents at 293 K.

Adsorbent	q_m_·(mg·g^−1^)	Reference
H-δ-MnO_2_ nanoparticles	427.0	MO 5
PANF-g-HPEI0.6	194.0	MO 7
PED-MIL-101	194.0	MO 8
BFSAP	167.0	MO 2
Activated porous carbon	97.1	MO 4
Protonated Chitosan	89.3	MO 1
surfactant modified silkworm exuvia	87.0	MO 6
Fe_2_O_3_-BC nanocomposite	20.5	MO 3
ADC	68.1	This work
PILDC	248.7	This work

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
