# Peer review of "The High Efficiency of Anionic Dye Removal Using Ce-Al13/Pillared Clay from Darbandikhan Natural Clay"

_molecules, 2019, doi:10.3390/molecules24152720_

Round 1

Reviewer 1 Report

In general, the readability of the manuscript should be increased by improving clarity and sentence structures. Relevance should be made more clear in introduction/conclusion; it is not directly clear why you are interested in clay. (You can find some suggestions below.)

Line 28 - Introduction

The introduction is somewhat incoherent and not so well structured. Now it is just a list of information that you , the paragraphs are not connected. And some sentences are quite complex/long and unclear. Please improve! Suggestion: bring the third paragraph (line 42-51) to the beginning/start of your introduction.

Line 37-line 41

Long sentences, please rephrase.

Line 53

Why are you specifically interested in MO?

Line 56-line 59

Please split into two sentences. (… of the medium. The results of …)

Line 63

Should be 2. Materials and methods (not 1. Materials and methods).

Line 64 – 2.1 Adsorbate

Please make complete sentences.

Line 65 A space is missing between 507 and nm.

Line 79

Either specify the type of centrifuge (radius) that was used, or express the centrifugal speed in g (relative centrifugal force). (This accounts for all mentioning of rpm.)

Line 131 Please introduce abbreviation CEC in full.

Line 158 – Table 1

What are these numbers: concentrations, masses … ? And in what unit?

Line 160

How do you know that the reduction of calcite content is the cause for the enrichment of SiO2 and Al2O3? Please specify or rephrase.

Line 162 – “this cause an increase in the surface acidity and porosity”

“cause” should be “caused”. Where are the data on surface acidity?

Line 177

The bands at 2876.10 and 2512.73 are not depicted, they are excluded from the graph by the two segmental x-axis. (in the gap between the two segments)

Line 187

Do you mean that there was no significant change between the three clay types?

Line 191

Please check this calculation; should it not be 12.0%?

Line 194/195  - Description figure 1

Either use the same color coding in each figure subset, or add the legend of figure 1c in the description as well.

Line 201 – “Compared to other pillared or acid activated clays …”

Which other pillared or acid activated clays?

Line 193 / line 204

Po (superscript) or po (subscript)? Be consistent.

Line 229 – “adsorptive”

Please be consistent. With the words adsorption/adsorbate/adsorbents already in there; please stick to these terms.

Line 249 – figure 2

Be consistent with the color coding; e.g. red for ADC and blue for PILDC. Do not switch this within two subsets of one figure.

Line 286 – Table 3

Please make this table more clear: e.g. by extending/adding lines from kinetics model (left column) all the way to the right column. Now there are two lines, below experimental and below R2 at pseudo-first-order? These make it confusing.

Line 295-308

Is there a specific reason to put this in the results section? Otherwise move to materials and methods section 2.4, it disturbs the fluency of you results section.

Line 318 – “The Freundlick KF and n values for the pillared clay are all higher than those of the acid activated clay at the same temperatures.”

This is not true, according to table 4: the KF values are indeed higher, but the n values are lower in PILDC (compared to ADC).

Line 321 – “The Langmuir adsorption capacity …”

Where are these data presented?

Line 331-332

The superscripts are inconsistent: number zero or letter o. (So/SO/S0)

Line 341-342

Can you try to explain the differences in ΔG between ADC and PILDC? At most temperatures ΔG is less negative, except at 303K. Why?

Author Response

Response to Reviewer 1

The authors are thankful for their positive and constructive reviews.

A revised version was prepared with changes marked in which all points raised by the reviewers were considered. Explanations of the changes are given below.

·       Line 28 Introduction

Response: The introduction have been improved according to the reviewers suggestions list below and inserting further details and explanations: -

1-     The third Paragraph (line 42-51 Old) moved to the start of the introduction (Lines 29-40).

2-     Inserting (Sorption is used not only for the removal of dyes, but also for numerous pollutants and proteins. Activated carbon, silicate materials, biomasses [3] and chitosan [4] are common efficient and low cost adsorbents [5]. The cost/efficiency ratio determines the manufacturing procedure of these adsorbents [6])

·       Line 37 -41 Long sentences, please rephrase. 

Response:  The sentence was separated to three sentences and rephrased and become Lines 50-53  (Improving thermal stability, adsorptive and catalytic properties of pillared clay is very important. One of the improvements was to incorporate different mixed-oxides to produce pillared clays with enhanced properties. Al-pillared clays, in which other metal cations are introduced, are often used [14][15].)

·       Line 53 Why are you specifically interested in MO?   

Response: Two sentences were inserted with a reference

(Line 55) (Natural clays are effective, cheap and abundant adsorbents, but show poor adsorption efficiency toward anionic dyes [16]).

 Line 60-61 (Methyl orange is widely used as a model anionic dye in adsorption and photocatalytic degradation studies.)

·       Line 56 -59 Please split into two sentences.

Response: The sentence was split to two sentences:

Line 62-63 (The adsorption capacity for MO is considerably affected by the surface functional groups and surface charge of the adsorbents, as well as by pH and temperature of the medium).

Line 63-65 (The values of these parameters point to a possible involvement of physical forces such as hydrogen bonding, Vander Waals and covalent chemical bonds in the adsorption process.)

·       Line 63 Should be 2. Materials and methods (not 1. Materials and methods).

Response: Corrected to: became Line 69 (2. Materials and methods)

·       Line 64 – 2.1 Adsorbate Please make complete sentences.

Response: Corrected:-  Line 71 (A model anionic dye, methyl orange (MO), was used as adsorbent (λmax at pH ≥ 3.1 = 507 nm, and at pH< 4.4 =618 nm).

·       Line 65 A space is missing between 507 and nm

Response: Corrected. Line 71 (A model anionic dye, methyl orange (MO), was used as adsorbent (λmax at pH ≥ 3.1 = 507 nm, and at pH< 4.4 =618 nm).

·       Line 79 Either specify the type of centrifuge (radius) that was used, or express the centrifugal speed in g (relative centrifugal force). (This accounts for all mentioning of rpm.)

Response: detail is given at Line 85-86 (The remaining solid was collected by centrifugation for 10 minutes (5000 rpm) at a relative centrifugal force (RCF)=4193 using Universal 320 centrifuge from Hettich GmbH & Co. KG,…..)

·       Line 131 Please introduce abbreviation CEC in full

Response: detail was provided at line 138-139 (The cation exchange capacity (CEC) was..).

·        Line 158 – Table 1 What are these numbers: concentrations, masses … ? And in what unit?

Response: Corrected, those are mass%, (%) was added to the oxides in Table 1 and they are unitless.

SiO2%

TiO2%

Al2O3%

Fe2O3%

MnO%

MgO%

CaO%

Na2O%

K2O%

P2O5%

SO3%

LOI

Sum

·       Line 160 How do you know that the reduction of calcite content is the cause for the enrichment of SiO2 and Al2O3? Please specify or rephrase.

Response: The sentence under Table 1 (Line 183-186) has been rephrased as below for more explanation:

Line 184-189 (The calcite content of DC has been reduced to a great extent from 21.4% CaCO3 (12.0% CaO) to 2.3% and 0.7% CaCO3 (1.3% and 0.4% CaO) for ADC and PILDC respectively by the action of the acid activation (dissolution) (Table 1) in addition to an increase in the porosity of ADC. The relative enrichment of SiO2 and Al2O3 may be related to the decrease in calcite content for ADC and PILDC. In the case of the PILDC, the ratio of the Al2O3 increased due to the insertion of the pillaring agent in addition to the effect of acid dissolution.)

·       Line 162 – “this cause an increase in the surface acidity and porosity”

“cause” should be “caused”. Where are the data on surface acidity?

Response: The phrase (this cause an increase in the surface acidity and porosity.) has been deleted.

·       Line 177 The bands at 2876.10 and 2512.73 are not depicted, they are excluded from the graph by the two segmental x-axis. (in the gap between the two segments)

Response: The band at 2512.73 is removed from line 177. And the sentence became (The bands observed at 2876.10, 1432, and 712.87 cm-1. correspond to carbonate)

Fig. 1.b is corrected for the X-axis at the left side which starts from 2800 cm-1 not 3300 cm-1 which include the band 2876.10.

·        Line 187 Do you mean that there was no significant change between the three clay types?

Response: No we do not mean that, we mean that the difference in weight loss in the temperature range (460-590) for the three clays are not big. But for the other temperature ranges the differences were much higher.

·       Line 191 Please check this calculation; should it not be 12.0%?

Response: 12.3% was wrong, the correct sentence became Line 217-218 (The 9.35% weight loss of CO2 corresponds to 21.3% total calcite content of the clay which is consistent with the presence of 12.0% of CaO (Table 1). The 12.0% of CaO corresponds to about 21.4% calcite content.)

·       Line 194/195 - Description figure 1 Either use the same color coding in each figure subset, or add the legend of figure 1c in the description as well.

Response: Figure 1 has been improved for the colors and description of figure 1c was added as Line 222-224 (Figure 1. (a) X-ray diffractogram for DC (blue), ADC (black) and PILDC (red), (b) FTIR Spectra (c) TG curves for DC (blue), ADC (red), PILDC (green) and (d) Adsorption-desorption isotherms of N2 at 77 K for DC, ADC, and PILDC.)

·       Line 201 – “Compared to other pillared or acid activated clays …” Which other pillared or acid activated clays?

Response: two references have been added for the comparison and the sentence became Line 230-232 (Compared to other pillared clays like Bentonite [36] or acid activated clays [37], however, the increase of the surface area was relatively low, which could be explained by the lower smectite content).

·       Line 193 / line 204 Po (superscript) or po (subscript)? Be consistent.

Response: Corrected to subscript and became Line 233 (above the adsorption branch at P/Po below about 0.4.)

·       Line 229 – “adsorptive” Please be consistent. With the words adsorption/adsorbate/adsorbents already in there; please stick to these terms.

Response: Corrected and became Line 258-259 (It affects both the charge of the adsorbate as well as the charge on the adsorbent surface)

·       Line 249 – figure 2 Be consistent with the color coding; e.g. red for ADC and blue for PILDC. Do not switch this within two subsets of one figure.

Response: Figure 2 corrected. Red color for ADC and blue color for PILDC used for all a, b, c, and d.

·       Line 286 – Table 3 Please make this table more clear: e.g. by extending/adding lines from kinetics model (left column) all the way to the right column. Now there are two lines, below experimental and below R2 at pseudo-first-order? These make it confusing.

Response: Table 3: the lines for the kinetic models were extended from left to right and another line was added for PILDC (Experimental).

·       Line 295-308 Is there a specific reason to put this in the results section? Otherwise move to materials and methods section 2.4, it disturbs the fluency of you results section.

Response: Lines 295-308 were moved to Section 2.4 Materials and Methods (Lines 155-171).

·       Line 318 – “The Freundlick KF and n values for the pillared clay are all higher than those of the acid activated clay at the same temperatures.” This is not true, according to table 4: the KF values are indeed higher, but the n values are lower in PILDC (compared to ADC).

Response: Corrected, (n) removed and the sentence became Line 336-337 (The Freundlich KF values for the pillared clay are higher than those of the acid activated clay at the same temperatures. This observation indicates that PILDC has better adsorption intensity for MO adsorption than ADC.)

·       Line 321 – “The Langmuir adsorption capacity …” Where are these data presented?

Response: The data for Langmuir isotherm were added to Table 4 which were estimated from the plots of Langmuir, Figure 4 also expanded to include Langmuir isotherm and the sentence (Line 308-311) was modified to Line 327-329 (The adsorption of MO on ADC (Figures 4.a & 4.c) and PILDC (Figures 4.b & 4.d) at different temperatures, shows good fitting of the isotherm models of Langmuir, Redlich-Peterson and Freundlich as depicted by the error functions R2 and SSE (Table 4).

·       Line 331-332 The superscripts are inconsistent: number zero or letter o. (So/SO/S0)

Response: The superscripts were corrected and united for:-

a-     Equations 10-12 at Lines 347-348.

b-     Corrected at Lines 346, 347, 349, 355, 358, 359

c-     Corrected for Table 5 at Line 354.

·       Line 341-342 Can you try to explain the differences in ΔG between ADC and PILDC? At most temperatures ΔG is less negative, except at 303K. Why?

Response:  The value of ΔG for PILDC (-286) at 303 K in Table 5 (at the far right column) was wrong, the point was missing and corrected to (-2.86).

Reviewer 2 Report

Manuscript Number: Molecules-538799

Dear authors,

In the present paper, the removal of anionic Dye from aqueous solutions was studied by using natural Darbandikhan clay as sorbent. The results presented by the authors could be of interest for wastewater treatment applications. The manuscript requires major corrections before publication:

1.       The introduction needs to be improved. It is not clear the relevance of using the sorption method for the removal of Dye. I recommend citing these recent works for supporting the introduction. Sorption may be used not only for the removal of dyes, but also for a big amount of pollutants and proteins (the process is cost-efficient):

·         , Environ Sci Pollut Res 24(1) (2017) 15-24.

·         Journal of Environmental Chemical Engineering 6(2018) 5351-5360

·         Chemical Engineering Journal 361(2019) 839-852

2.       Please explain with more details, how the authors have obtained the constants of Redlich-Peterson by using the non-linearized form isotherm?

3.       If the process is supposed to be dominated by physical sorption, why the kinetic data is better fitting with pseudo-second order model?

4.       Please provide a summary table for comparing the sorption capacity (qmax) of the sorbent materials and the results reported in the literature (for dye removal).

5.       What about the desorption studies? How could be regenerated the sorbents after exhaustion?

Author Response

Response to Reviewer 2 Comments  

The authors are thankful for their positive and constructive reviews.

A revised version was prepared with changes marked in which all points raised by the reviewers were considered. Explanations of the changes are given below.

1-     The introduction needs to be improved. It is not clear the relevance of using the sorption method for the removal of Dye. I recommend citing these recent works for supporting the introduction. Sorption may be used not only for the removal of dyes, but also for a big amount of pollutants and proteins (the process is cost-efficient):

 Environ Sci Pollut Res 24(1) (2017) 15-24.

· Journal of Environmental Chemical Engineering 6(2018) 5351-5360

· Chemical Engineering Journal 361(2019) 839-852

Response: The introduction has been improved as following:

a-     The third Paragraph (line 42-51) moved to the start of the introduction (Lines 29-40).

b-     The suggested references were added (Line 35-38) (Sorption may be used not only for the removal of dyes, but also for a big amount of pollutants and proteins. Activated carbon, silicate materials, biomasses [3] and chitosan [4] are common efficient and low cost adsorbents [5]. The cost/efficiency ratio determines the manufacturing procedure of these adsorbents [6])

c-     The sentence at Lines 37-41 was separated to three sentences and rephrased and become Lines 50-53  (Improving thermal stability, adsorptive and catalytic properties of pillared clay are very important. One of the improving methods was to incorporate different mixed-oxide to produce pillared clays with enhanced properties. Mostly Al-pillared clays in which other metal cations are introduced [14][15].)

d-     The sentence at Line 29 (Clays are considered potential adsorbents for environmental purposes [1]) was deleted.

Line 52-55 rephrased to Lines 55-60 (Natural clays are effective, cheap and abundant adsorbents, but show poor adsorption efficiency toward anionic dyes [16]. While some clay minerals and their acid treated or modified forms, have been recognized for adsorption capacity for methyl orange (MO), among them; Moroccan natural clays [17], heat-treated palygorskite clays [18], intercalation of cetyltrimethylammonium bromide to vermiculite interlayer spacing [19], and protonated cross-linked chitosan [20]. Methyl orange is widely used as a model anionic dye in adsorption and photocatalytic degradation studies.

2-     Please explain with more details, how the authors have obtained the constants of Redlich-Peterson by using the non-linearized form isotherm?

Response: Several computer softwares can be used for non-linear curve fitting like MATLAB, OriginPro or Solver in Excel sheets. For the current work, OriginPro was used. Line 166-168 {Redlich-Peterson (R-P) isotherm (Eq. 9) was applied as a three parameter isotherm using non-linear curve fitting with the aid of OriginPro computer software with Levenberg Marquardt iteration algorithm}.

3-     If the process is supposed to be dominated by physical sorption, why the kinetic data is better fitting with pseudo-second order model?

Response: Before Figure 3, a sentence was added: Line 313-314 (However, the distinction between physical and chemical adsorption is difficult because no sharp distinction exists and intermediate cases are present.)

4-     Please provide a summary table for comparing the sorption capacity (qmax) of the sorbent materials and the results reported in the literature (for dye removal).

Response: Table 6 was added (Line 368) to the manuscript for comparison of the current adsorbents to that of literature.

5-     What about the desorption studies? How could be regenerated the sorbents after exhaustion?

Response: The desorption and regeneration will be our next project, it has been added to the manuscript at the end of the Conclusion as Line 381-382: (Desorption, regeneration and column (continuous system) are to be considered for the future work to bring the current work to applied world).

Round 2

Reviewer 1 Report

The authors have sufficiently revised the manuscript and the paper is ready for publication now.

Reviewer 2 Report

Dear Editor,

The manuscript has been improved. It can be accepted after minor revisions:

I recommend to the authors to take care of the references format.

The references [3], [4], [5], [6], etc.  do not follow the  format of molecules journal. Please add the corrected format